# Recognition of Potential Geosites Utilizing a Hydrological Model within Qualitative–Quantitative Assessment of Geodiversity in the Manawatu River Catchment, New Zealand

**Vladyslav Zakharovskyi [1] and Károly Németh [1,2,3,4,5,*]**

1 School of Agriculture and Environment, Massey University, Palmerston North 4474, New Zealand
2 Lithosphere Physics Group, Institute of Earth Physics and Space Science, 9400 Sopron, Hungary
3 The Geoconservation Trust Aotearoa Pacific, Ōpōtiki 3122, New Zealand
4 National Earthquakes and Volcanoes Program, Saudi Geological Survey, Jeddah 21514, Saudi Arabia
5 Istituto Nazionale di Geofisica e Vulcanologia, 40128 Bologna, Italy
* Correspondence: k.nemeth@massey.ac.nz; Tel.: +966-56-763-4717

**Abstract:** Hydrology is one of the most influential elements of geodiversity, where geology and geomorphology stand as the main values of abiotic nature. Hydrological erosion created by river systems destructing rock formations (eluvial process) from streams' sources and then transporting and redepositing (alluvial process) the rock debris into the main river channels, make it an ongoing transformation element of the abiotic environment along channel networks. Hence, this manuscript demonstrates the influence of hydrological elements on geosite recognition, specifically for qualitative–quantitative assessment of geodiversity, which is based on a combination of geological and geomorphological values. In this concept, a stream system will be treated as an additional element. The basement area of the Manawatu Region has been utilized as the territory for the research of hydrological assessment. The region is in the southern part of the North Island of New Zealand and has relatively low geological and geomorphological values and diversity. The Strahler order parameter will be demonstrated as a hydrological element for geodiversity assessment. This parameter has been chosen as one of the most common and acceptable within geographical information system (GIS) environments. The result of this assessment compares the influences of Strahler order on qualitative–quantitative assessment of geodiversity and provides its drawbacks. Additionally, the places with high values will be considered for more accurate field observation to be nominated as potential geosites with an opportunity for geoeducational and geotouristic significance.

**Keywords:** Strahler order; river system; QGIS; geoeducation; geotourism; Cenozoic; Mesozoic; fluvial; sand; gravel

## 1. Introduction

The description and evaluation of abiotic nature for geosites and geopark establishments are currently the main goals of geodiversity assessments. Accurate geosite recognition will help a researcher to minimize the area of research and concentrate on the places with specific features valuable for geotourism [1] and geoeducation [2–6]. They are the main ways to increase the educational level of students and tourists about the processes forming the abiotic environment such as volcanoes, sedimentary basins, soils, climate, eluviation and denudation, chemical (and biological) erosion, and solar insolation [7–13]. All these processes are always operating continuously, forming, and transforming the geological and geomorphological parameters of the Earth's surface. Here, geology and geomorphology as elements of geodiversity must be considered as the basement of the non-living environment, whereas other elements are its transforming agents (e.g., climate) or remnants (e.g., soils) of the rock formations. Currently, scholars include in the term of geodiversity the following elements: geology, geomorphology, hydrology, climate, soils,

space energy (meteorites, gravitation, and solar insolation), tectonic processes, and biotic and anthropogenic influences [3,4,14]. Hence, understanding of the geological and geomorphological parameters help to create a general view on the surface, whereas other elements of the abiotic nature will describe the process of the rock cycle, where hydrology is becoming one of the most influential [15–17].

Hydrology within geodiversity is a special erosional element, simultaneously filling depressions on the surface and transforming and accumulating sediments. It links hydrology directly to geological elements in geodiversity description. Hence, the assessment of hydrological parts from a geodiversity perspective have been studied for the number of different locations to create geodiversity model. Some research has been concentrated on specific parameters such as rock fractures and their permeability, together with geomorphology and aquifer features [18,19], whereas others have focused more on the water physical–chemical properties [20]. Furthermore, study of a waterfall demonstrates the importance of its hydrological features from cultural [21], esthetic [21,22], scientific [21], economic [23], and touristic [22–24] perspectives. Standard maps show hydrological elements through objects such as lakes, streams, rivers, marshlands, and others. These elements influence geological formations with eluvial and alluvial processes especially streams and rivers as active water flows [25]. These processes start from the streams' sources (springs and underground water channels), coursing from the mountain areas to reach lowlands and depressions to achieve an equilibrium state, which is mostly presented with the marine basin supplying it with circulation, sediments, and nutrition [26]. River flows form valleys through their power to cut the surface of rock formations and transport its material downflow [27]. This process, most of the time, creates a dichotomic merge of streams from various sources passing through diverse catchment areas. The merging flow generates a geologically more diverse and complex riverbed with an increasing variety of rock fragments to display transported material along the river source to its mouth [28]. Hence to emphasize the influences of a river system on geodiversity, this research utilizes the Strahler order, which expresses the rank of each channel, where the order grows with two channels (same order) merging into new within the assessed catchment area. Together with qualitative–quantitative assessment of geodiversity, the Strahler order will characterize the part of the studied river, which is likely to a contain high amount and diverse array of transported rock material (sediments). Then, it can be used as a proxy for geodiversity (sediment variety) along the stream networks from source to sink.

Qualitative–quantitative assessment of geodiversity (QQG) has been developed for recognition of locations with potentially high geodiversity based on the accessible (open) geospatial database (e.g., SRTM, geological and topographical maps) and simple methodology [29]. Geology and geomorphology are the two main elements represented by the general state of the abiotic environment, whereas this research also add the hydrological parameter into the equation to study its influence on the model [5]. The Strahler order is a standard hydrological parameter, which can be calculated from any digital elevation model utilizing common geographical information systems (GIS) software (e.g., QGIS, ArcGIS, Grass GIS, Saga GIS). Hence, this parameter can be included into QQG without changes to the methodological goal to make the assessment applicable for any territory throughout the globe, making it acceptable for every researcher regardless their level of knowledge of the GIS software.

The aim of this manuscript is to include the hydrological element into QQG methodology to test the locations with low geological values from the global perspective. Our working hypothesis is that the result of the assessment will show that the level of information from the hydrological element can have an influence on the general geodiversity values (geological and geomorphological elements) while recognizing the pitfalls of the hydrological modeling. The area of research is the catchment area of the Manawatu River in the lower North Island of New Zealand, which is geologically represented by Mesozoic greywacke and various post-Miocene siliciclastic rocks on the surface and their geological variety. Meanwhile, the additional goal of this work is to identify places with potential

locations acceptable for further, more accurate description and establishment of geosites as places with high geoeducational and geotouristic values.

The manuscript identified two knowledge gaps we aimed to explore in this research. One is more global, whereas the second is more regional in relevance. The global knowledge gap is that we have very limited knowledge on how hydrology, river characterization, and catchment area investigation can contribute to the overall geoheritage valorization and geodiversity estimates. This is since only a handful of studies have addressed the significance of rivers in geoheritage works and most of them approached the problem in a very general way or used the rivers just as a link between otherwise important geological and geomorphological sites mostly along their aesthetic values. In this paper, we identify this is a knowledge gap and we intended to explore this.

On other hand, the Manawatu River is a main geomorphological element, the symbol of an entire region in the lower North Island of New Zealand and commonly appears in geoconservation strategies as a key element for nature conservation and future geotouristic works. Although we see this as a promising starting point to initiate such ventures, we identified a significant gap between this plan and the conducted or planned research to explore the real weight of the Manawatu River in geoheritage valorization and geodiversity estimates. In this paper we addressed this issue as well.

## 2. Overview of Manawatu Basin

Manawatu Basin is in the south part of New Zealand's North Island (Figure 1). Its area is 5850 km$^2$, which includes three NE–SW trending mountain ranges: Tararua in the south, Ruahine in the north, and Waewaepa in the east [30,31]. The Tararua and Ruahine ranges are dissected by the Manawatu River in the central part, forming the Manawatu Gorge. Meanwhile, the east side of the axial ranges is typical rolling hill country, as part of the folded and faulted accretionary prism formed in front of the obliquely westward subducting Pacific Plate beneath the Indo-Australian plate [32,33]. The western side of the range is a broad coastal plain with spectacular marine and river terraces, recording the rapid uplift of the region in the last million years [34,35].

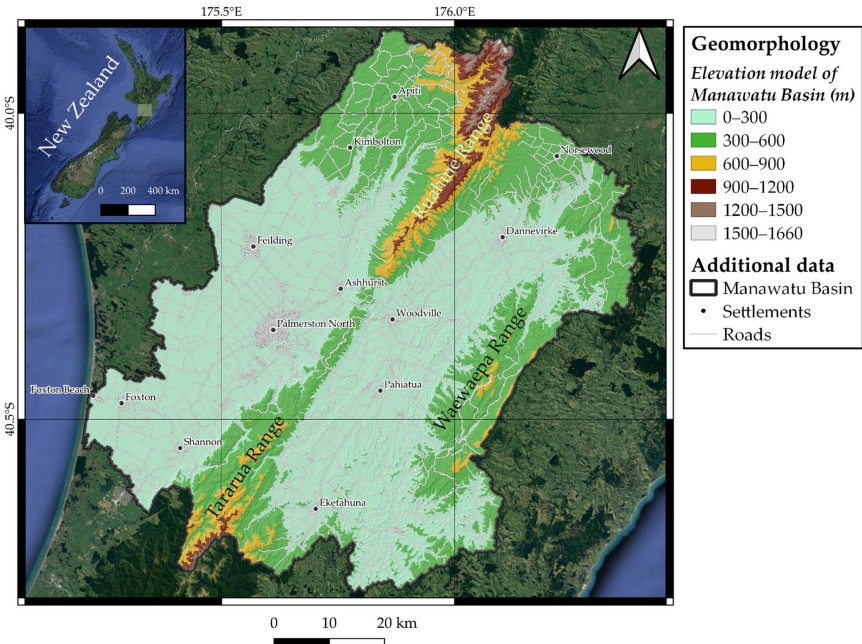

**Figure 1.** Overview map and elevation model of Manawatu Basin created from Shuttle Radar Topography Mission (SRTM) 1 Arc-Second Global (https://www.usgs.gov/centers/eros/science/usgs-eros-archive-digital-elevation-shuttle-radar-topography-mission-srtm-1, accessed on 15 December 2022); background is Google terrain map. The coordinate system is WGS 84 (EPSG: 4327), the same in all other figures.

*2.1. Geology and Geomorphology*

The geological concept of the Manawatu Basin is formed by a range of different sediment formations from Jurassic to Holocene periods, which are remnants of the active Manawatu River system. Geological data presented for the Manawatu Basin has been extracted from a 1:250,000 scale New Zealand geological map (Q-Map Series—https://www.gns.cri.nz/data-and-resources/geological-map-of-new-zealand/, accessed on 23 December 2022) [36] (Figure 2). Uniquely for the region, in the north part of the Tararua Range exposed in the Manawatu Gorge area, the rocks of the Kaweka Terrane formation present with Jurassic Basalt. In general, the geological description of the Manawatu Basin is described with 32 different rock groups, which we decided to range according to their ages, where the oldest are from Mesozoic era (black and deep purple colors) (Figure 2), including mostly siliciclastic sedimentary rocks from the Jurassic period grouped into tectonostratigraphic units such as Torlesse Composite Terrane, Pahau Terrane, Rakaia Terrane, and Kaweka Terrane and the Cretaceous period presenting with Mangapurupuru Group and Tinui Group. Mesozoic rocks are dominated by greywacke, which is the most common basement rock type forming mountain ranges through the whole North Island of New Zealand. Here, greywacke as the main lithology of the axial ranges and has been tilted and forms the mountain range dissected by the Manawatu Gorge. The Tararua Range in the south has an altitude 200–1300 m that increases towards the south, whereas the Ruahine Range on the contrary grows to the north more rapidly to 1000 m height and then reaches to 1500 m. Moreover, there are some additional older rocks forming ranges on the east side of the main axial ranges such as the Waewaepa Range reaching up to 700 m above sea level. Then, there are Cretaceous rocks represented by mudstone and sandstone mostly cropping out in the northwest and covering the smallest surface area compared with other older Mesozoic rock groups. The whole west part of the Manawatu River catchment area as well as the base of the Manawatu Gorge and most of the land on the north covered by Miocene–Pliocene deposits of shallow marine sedimentary rocks (pink color) representing the history of an evolving accretionary prism along the convergent plate margin. These rocks were formally included in 15 lithostratigraphy groups such as the Hurupi Group, Makurim Group, Mangaheia Group, Mangamaire Group, Mangatu Group, Maxwell Group, Moa, Napier Group, Onoke Group, Pakihi Supergroup, Palliser Group, Soren Group, Te Aute, Te Hoe Group, and Tolaga Group. These areas are rolling hills today with heights from 200 up to 700 m. The largest area of the catchment is constructed with the youngest Pleistocene–Holocene sediment groups (orange color), where Late Pleistocene sediments include the Kai-Iwi Group, Kidnappers Group, Okehu Group, Shakespeare Group, and Middle Pleistocene sediments and various Late Pleistocene, Middle Pleistocene, and Early Pleistocene sediments consist of mud, silt, sand, and gravel from shallow marine to fluviolacustrine origin. These areas are mostly flat-topped, hosting spectacular marine and fluvial terraces along the Manawatu River and small stream valleys draining from the north. The last formation is Late Pleistocene to Holocene periods (yellow color), which have been merged into a single category in our model as they are the modern alluvial and riverbeds. Alluvial deposits are formed by three groups including Holocene sediments, Pleistocene–Holocene sediments, and Late Pleistocene–Holocene sediments. Their present-day altitude ranges from 0 to 100 m above sea level coming with riverbeds from the west, north-west, and north parts of the Manawatu Basin and transported to the south-west forming a large plain area from Palmerston North to Foxton Beach area, where the Manawatu River enters the Tasman Sea. Hence, the geological history of the Manawatu region is locked into 31 sedimentary and 1 igneous (basaltic) rock types formed since the Jurassic period and eroded through the Manawatu River system.

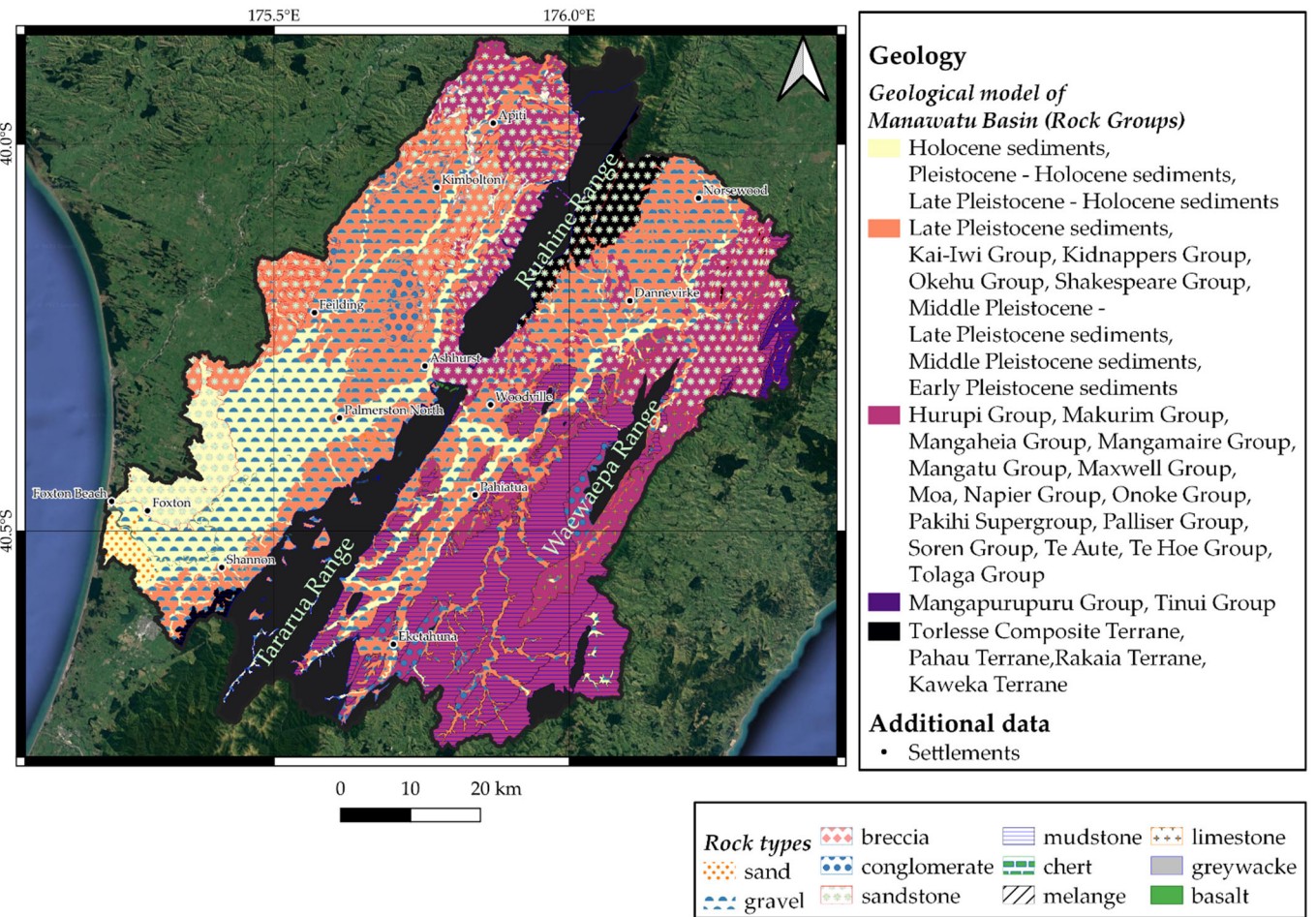

**Figure 2.** Geological model of Manawatu Basin based at the 1:250,000 scale, New Zealand geological map (Q-Map Series—https://www.gns.cri.nz/data-and-resources/geological-map-of-new-zealand/, accessed on 23 December 2022) [36]; background is Google terrain map.

### 2.2. Hydrological System and Climate

The hydrological system of the Manawatu Basin presents with 160 streams and 46 creeks that fall into 13 rivers and 1688 lakes according to the data from the 1:50,000 topographic map of New Zealand and downloaded from Land Information New Zealand (LINZ) (https://data.linz.govt.nz/layer/50327-nz-river-centrelines-topo-150k/, accessed on 26 December 2022). Streams and creeks mostly come from the three mountain ranges and their surroundings captured into the basin area (Figure 3): Ruahine (north), Waewaepa (west), and Tararua (south) [30,31]. They supply 13 rivers, where 10 of them flow from different sources in the central and the eastern part of the Manawatu Basin and merge near Manawatu Gorge. Then, the formed river cut through and gradually meandering to the south and fall in Tasman Sea. The upper part of the Manawatu River is sourced from the Ruahine Range from its east side (Figure 3) and flows to the south, where the Mangator and Taimaki Rivers merge with the Manawatu from the Waewaepa and Ruahine Ranges, respectively. Then, the river turns toward the west in the center of the Gorge, where it has the Tiraumea, Makakahi, and Mangahao Rivers as its main inflows. Meanwhile, the Tiraumea River supplies the Mangaone and Ihuraua Rivers from the south and the Makuri from the east. There is a similar situation with Makakahi River, which supplies the Mangatainoka River sourced from the Tararua Range. The lower part of the Manawatu River starts from the Manawatu Gorge and continues towards the south up to Foxton Beach, where its mouth reaches Tasman Sea. In the lower Manawatu, three small rivers supply its flow. Pohangina and Oroua both come from the west slopes of the Ruahine Range, whereas the Tokomaru comes from the west slopes of the Tararua Range. Moreover,

the Manawatu Basin contains 1688 lakes (average size 3198 m²) spread through the whole area of research, mostly concentrated in the western and southeastern parts. From them, three reservoirs are in the south part of the Tararua Range and the Karere Lagoon is located near the lower flow of the Manawatu River. Finally, two other named lakes are located in the northeastern part of the basin: Mahangaiti and Rotoataha. Hence, the hydrological element of Manawatu Basin is formed by the Manawatu River sourced from the Ruahine Range that flows through Manawatu Gorge and leads more to the south, where it falls into Tasman Sea [37]. On its way it supplies 12 rivers and high number of creeks and streams. Furthermore, the region has 1688 lakes.

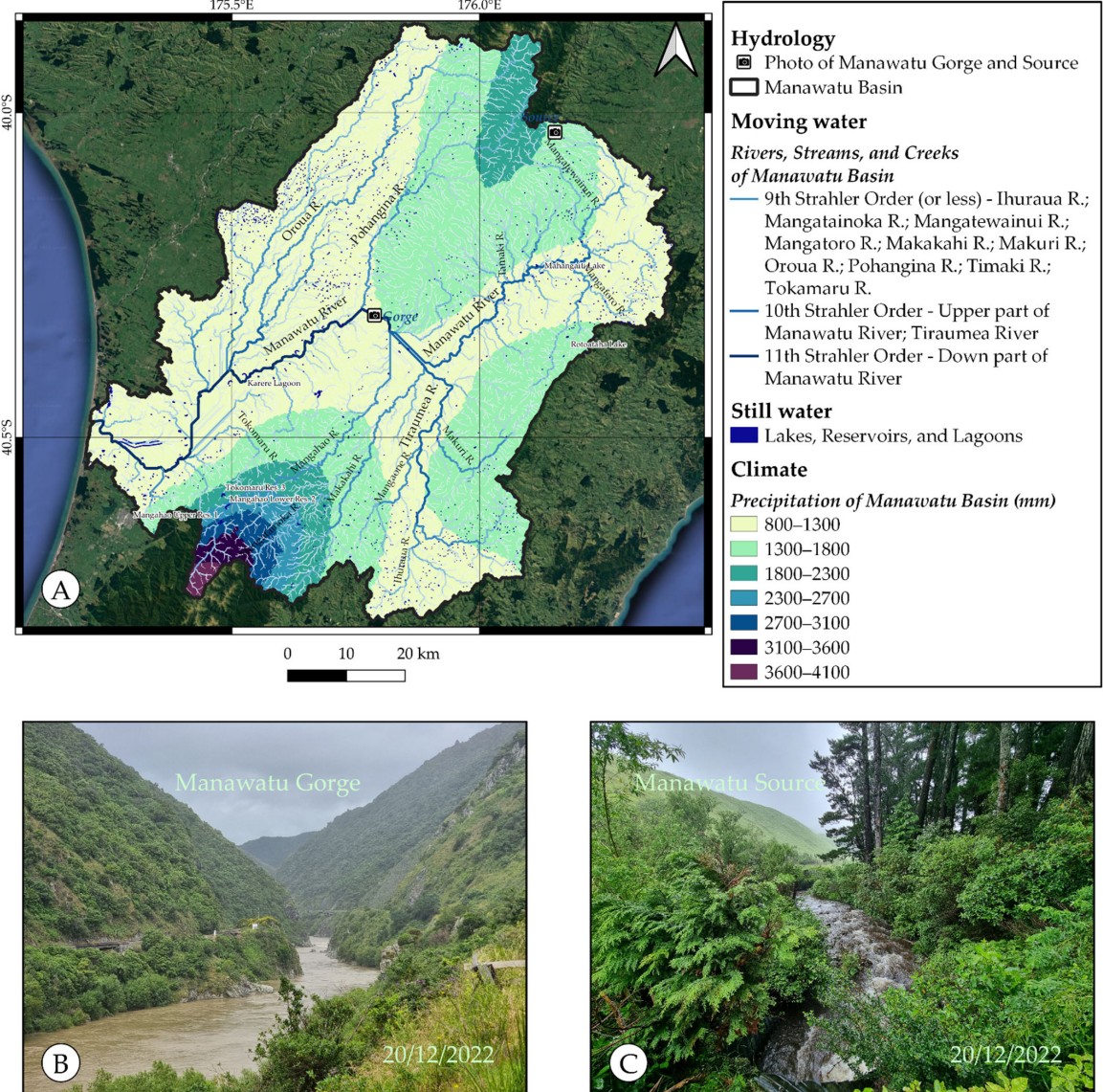

**Figure 3.** (**A**) Hydrological model of Manawatu Basin based on channels model extracted from Shuttle Radar Topography Mission (SRTM) 1 Arc-Second Global (https://www.usgs.gov/centers/eros/science/usgs-eros-archive-digital-elevation-shuttle-radar-topography-mission-srtm-1, accessed on 15 December 2022); the annual rainfall model downloaded from the site of Ministry for the Environment (https://data.mfe.govt.nz/data/category/environmental-reporting/atmosphere-climate/precipitation/global/oceania/new-zealand/, accessed on 26 December 2022); Lakes extracted from Land Information New Zealand (LINZ) (https://data.linz.govt.nz/layer/50293-nz-lake-polygons-topo-150k/, accessed on 26 December 2022); background is Google terrain map. Pictures of (**B**) Manawatu Gorge and (**C**) Manawatu source.

The climate of Manawatu Basin has been included into the description to show how precipitation supplies the region. The precipitation data have been downloaded from the site for the Ministry for the Environment (https://data.mfe.govt.nz/data/category/environmental-reporting/atmosphere-climate/precipitation/global/oceania/new-zealand/, accessed on 26 December 2022). The model demonstrates the average annual rainfall in Manawatu Basin. The flat areas and low hills formed under the biggest rivers, such as Manawatu, Tiraumea, Oroua, and Pohangina, contain annually around from 800 to 1300 mm of rainfall. Hilly areas around the mountain ranges have a higher amount of precipitation, from 1300 to 1800, where it rises higher closer to higher altitudes, reaching 1800–2300 in the Ruahine Range. Meanwhile, the Tararua Range precipitation rate keeps rapidly increasing towards the south, reaching around 4100 mm per year. Hence, the annual rainfall conditions of the Manawatu region produce a high amount of water, which keeps suppling the creeks, streams, and rivers of the basin. For example, two photographs have been taken near Palmerston North to demonstrate the transformation of the lower Manawatu after a week of rainfall (Figure 4).

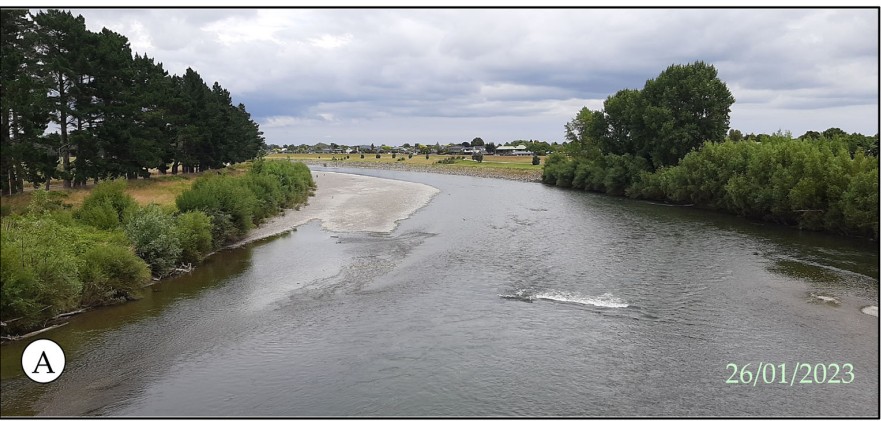

Lower part of Manawatu River near Palmerston North (condition after week with precipitation rate lower then 2.7 mm per day before photo was taken).

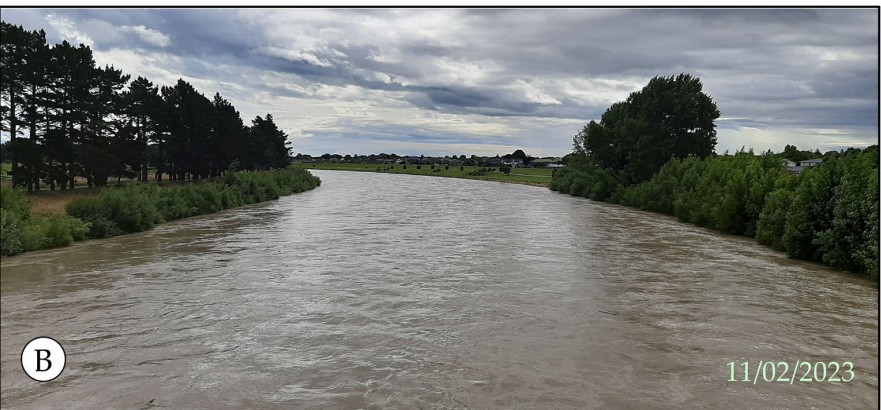

Lower part of Manawatu River near Palmerston North (condition after week with precipitation rate higher than 2.1 mm with two days 5.5 mm and peak in 11 mm few days before photo was taken).

**Figure 4.** Precipitation influences on the Lower Manawatu River near Palmerston North. (**A**) He Ara Kotahi Lookout during warm weather. (**B**) He Ara Kotahi Lookout after a week of precipitation. (**C**) Overview map of location, where photos have been taken.

## 3. Methodology

### 3.1. Assessment of Geodiversity

The methodology for assessment of geodiversity is currently versatile as researchers have their view on the range of abiotic elements that must be calculated. An important role in the assessment is the aim of the research, which Gray (2005) has been describing 31 of geodiversity values [38], starting from scientific and economic aspects to historical and spiritual values (e.g., geocultural values). Moreover, also important are issues with the accessibility of data (e.g., accurate geological data for the Samoa Islands [10]) and software (e.g., ArcGIS) for specific areas of research, as well as the GIS knowledge of the scientists. Therefore, to avoid these issues we utilized the free access QGIS (3.16 "Hannover") (https://qgis.org/en/site/forusers/download.html, accessed on 12 December 2022) software, with its plugin "SRTM-Downloader" (https://plugins.qgis.org/ plugins/SRTM-Downloader/, accessed on 15 December 2022), which allows download of the 30 m resolution Shuttle Radar Topography Mission (SRTM) 1 Arc-Second Global model (https://www.usgs.gov/centers/eros/science/usgs-eros-archive-digital-elevation-shuttle-radar-topography-mission-srtm-1, accessed on 15 December 2022). Additionally, Zwoliński (2018) describes more about three types of geodiversity assessment [6] such as (1) qualitative—based on expert knowledge [39,40], (2) quantitative—based on the amount and accuracy of the raw data [9,11,41], and (3) qualitative–quantitative—where less accurate raw data is evaluated with an expert view [10,29,42]. Therefore, we decided to build our methodology on a qualitative–quantitative model utilizing basic geological data (rock type and age) combined with SRTM for geomorphological calculations. As a result, we have been developing qualitative–quantitative assessment of geodiversity (later in text QQG) with the aim to highlight places with possible locations of geosites applicable throughout the globe.

A geosite is a specific location an in abiotic environment that contains information about geodiversity and surface evolution. QQG methodology is based on calculation of geodiversity elements that have been divided into main and additional values [7–9,13]. The main values of geodiversity are based on geology and geomorphology, as they describe rock formations, which is the core of abiotic nature, whereas other elements are influencing, transforming or altering material on the surface [5,10,42]. These elements of transformations are additional values including hydrology, climate, tectonics and volcanism, biological, and anthropological footprints. Therefore, QQG methodology assesses geological and geomorphological elements according to an 8-point evaluation system.

### 3.2. Evaluation System

The evaluation system is qualitative part of QQG, which ranges all abiotic features to separate important from less informative objects or processes. In the next section, we describe the evaluation system for geological, geomorphological, and hydrological elements of geodiversity.

#### 3.2.1. Main Values

Methodology of assessment of geodiversity utilizing the qualitative–quantitative model is based on calculations of several elements of abiotic nature, previously evaluated according to expert view (Table 1). Here, we present an 8-point evaluation system that has been developed for the main elements of geodiversity: geology and geomorphology. The geological evaluation system is developed around the rareness of rock formations exposed on the surface, which have been studied by Blatt, H. and Jones, R. L. "Proportions of exposed igneous, metamorphic, and sedimentary rocks" [43]. The result of his research is presented in percentage of different rock formations exposed on the surface, where sedimentary rocks are the most common (66%), metamorphic only from Precambrian era covers 17%, and extrusive and intrusive are 8% and 9%, respectively. Hence, all rock types divided with different eras have been transferred onto an 8-point evaluation system, where 1 is the lowest value containing all sedimentary Cenozoic formations, 2 (low value)—

sedimentary Mesozoic, 3 (low to middle value)—sedimentary Paleozoic, and 4 (middle value) became metamorphic from the Precambrian era. The middle to high value rocks are much rarer rock types as they cover areas less than 6% of the total: value 5—intrusive Precambrian, value 6 high—extrusive Cenozoic, and value 7—Mesozoic. Meanwhile, value 8 includes all of the rest of the rock formations as their areas cover 1% or less on the Earth's surface. Hence, the 8-point evaluation system is concentrated on all presented rock types exposed on the surface, making this method globally accessible and comparable with different territories throughout the world.

**Table 1.** The 8-point value systems for geodiversity assessment with hydrological element.

| Values (8-point system) | Main Values of Geodiversity | | Additional Value |
|---|---|---|---|
| | Elements of Geodiversity | | |
| | Geomorphology | Geology | Hydrology |
| | Slope | Rock type and ages | Strahler order |
| 1 (the lowest) | 0–11.25 | Sedimentary Cenozoic | Non required |
| 2 (low) | 11.25–22.5 | Sedimentary Mesozoic | |
| 3 (low to middle) | 22.5–33.75 | Sedimentary Paleozoic | |
| 4 (middle) | 33.75–45 | Metamorphic Precambrian | |
| 5 (middle to high) | 45–56.25 | Intrusive Precambrian | |
| 6 (high) | 56.25–67.5 | Extrusive Cenozoic | |
| 7 (the highest) | 67.5–78.75 | Extrusive Mesozoic | |
| 8 (the rarest) | 78.75–90 | Sed. (Precambrian), Met. and Intr. (Cenozoic, Mesozoic, Paleozoic), Extr. (Paleozoic, Precambrian) | |

The geological element describes the parameters of rock formations, whereas the geomorphological element shows the forms that these formations present after the historical pressures of endogenic (tectonic and volcanic) and exogenic (weathering, erosion, and alteration) processes have been constantly changing the surface, known also as the geological–geographical cycle [44–46]. However, geomorphological data mostly provide information about elevations of each point on a coordinate net. This information can be transformed and presented in a range of different parameters such as ruggedness, roughness, slope, aspect, or even something more complicated such as geomorphon, topographic position index (TPI), etc. Hence, the right model has to be chosen, which can help to enrich the aim of the research, minimizes the areas of field observations, and highlights the locations that are likely to have an outcrop.

In our previous work, "Geomorphological Model Comparison for Geosites, Utilizing Qualitative–Quantitative Assessment of Geodiversity, Coromandel Peninsula, New Zealand" [29], we studied this issue. Six different models have been compared between each other: slope, ruggedness, roughness, geomorphon, TPI, and total curvature. The slope model presents the degree of slope angle [47], whereas the ruggedness is the surface heterogeneity [48] and roughness is the rate of surface irregularity [49]. Total curvature is a combination of plan and profile curvatures [50], whereas the geomorphon [51,52] and TPI [50] are more complex models of a landscape that forms from depressions and valleys up to ridges and peaks. Both models are based on calculation of the central pixel in comparison to its neighbors. The result of geomorphological research shows that slope, ruggedness, roughness, and TPI give similar results and can be exchangeable between each other, whereas geomorphon and total curvature are inappropriate.

Therefore, the slope model has been evaluated with an 8-point system to make it a similar value to the geological one chosen for this manuscript. Furthermore, slope degrees more than 45 degrees show that these areas are likely to be free from all loose material according to the angle of repose. It shows a critical angle for a pile of material, which can be held without sliding [53]. Hence, we consider this law in an opposite way, the places with degrees higher than 45 are likely to be presented with hard rock or its loose material has been fixed with some different material such as vegetation. Hence, 45-degree angles are considered as a threshold between low and high values for geomorphology. However, we are not going to exclude all angles lower than the threshold, as often some rock formations can be found lying on the Earth's surface. In this situation the geological parameter will outweigh the low angle, as most sediments and metamorphic and volcanic rocks are high. The results show (Table 1) that the slope with the lowest values are areas with degrees less than 11.25—1. Then value 2—from 11.25 to 22.5 degrees, 3 points are (low to middle values) 22.5–33.75 degrees, and middle values have 4 points presented as slopes between 33.75 to 45. Then, values higher than 4 are considered more valuable, as they skip the threshold and are likely to expose an outcrop. The middle to high range is 5 points, which is 45–56.25 degrees; the high value is 6—56.25–67.5 degrees, and 7 points is 67.5–78.75 and represents the highest values. Finally, the rarest areas are presented by only some mountain areas and coastal cliffs with degrees 78.75 to 90. Hence, the evaluation system for the slope model has been tailored for global recognition, making it also like the geological model, acceptable for assessment for any territory throughout the world.

### 3.2.2. Additional Values

For additional values of geodiversity, this manuscript describes the study of hydrological elements of geodiversity. However, more accurate is the channel network presented by rivers, streams, creeks, and formed valleys with high amount of precipitation. This is one of the active parts of the hydrological element alongside marine processes, which are not considered in this research. The continuous activity of river systems provides various levels of erosion and transportation of sediments, which is valuable knowledge for understanding the surface transformation of hard rocks to sediments and sedimentary rocks. The riverbanks contain rock material from streams that enter the main river in the basin, providing a general lithological overview of the region's geology. Additionally, anthropological, and cultural aspects can be considered along riverbanks as they are mostly selected by humans for settlements, fishing, and hunting. However, GIS does not provide many tools for the assessment of channel networks. The Strahler order is one of the simple types of assessment, which, considering possible places of water sources in the valleys or at least locations, is where precipitation can provide an additional temporal water source. Therefore, as a result the model will demonstrate a theoretical stream network, which also covers the real network. Then, all theoretical water sources are considered as the first order, so when two 1st streams merge into one, the new stream becomes second order. The same process happens with two merged 2nd order streams, which creates 3rd order streams, whereas 1st order streams that fall into 3rd order streams do not influence the order number of the latter (Figure 5). Hence, this model provides good information about channels that are merging and carrying sediments from different sources, which is not a required evaluation as they already provide the correct value for QQG assessment. However, in this research, hydrology is considered as local diversity and its influence will highly depend on the territory of research.

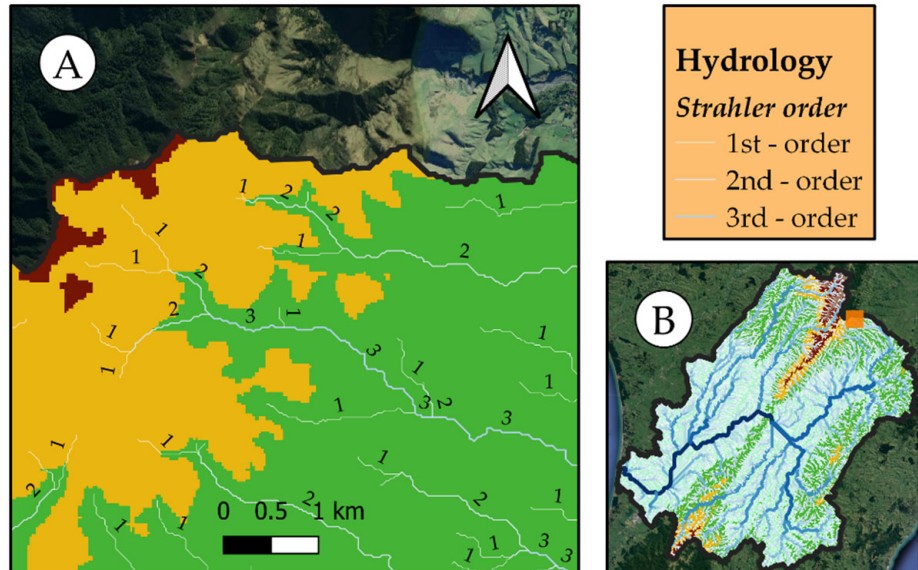

**Figure 5.** (**A**) Demonstration of Strahler order methodology (numbers are showing its stream order). (**B**) Overview map. The channel network calculated from SRTM model.

## 4. Results

The assessment of the Manawatu Basin has been completed in QGIS software utilizing the "Zonal statistic" tool, where a square grid with a 6.25 km$^2$ size for each cell was created. Then, the natural breaks (Jenks) [54] mode is used to group the final parameters into areas.

The general geodiversity model of the Manawatu Basin is based on the multiplication of two main values: geology and geomorphology. Geological data have been extracted from a 1:250,000 scale New Zealand geological map (Q-Map Series—https://www.gns.cri.nz/data-and-resources/geological-map-of-new-zealand/, accessed on 23 December 2022) [36] with next its evaluation according to an 8-point system presented in Section 3.2. Meanwhile, geomorphological data based on SRTM (Shuttle Reader Topography Mission) [55] and transformed into the slope model utilizing "Slope, Aspect, Curvature" of "Terrain Analysis-Morphometry" tool of Saga GIS implanted into QGIS software. The calculation is based on default method "9 parameter 2-nd order polynom" created by Zevenbergen and Thorne (1987) (https://onlinelibrary-wiley-com.ezproxy.massey.ac.nz/doi/epdf/10.1002/esp.3290120107, accessed on 19 December 2022) [38].

The result of the multiplication of geological and geomorphological data presents a range from 1 to 35 (Figure 6), where natural breaks mode divided it on five categories. The first is 1–2 and is the lowest vales of geodiversity; these territories cover a quarter of the southwest and less in the central northeastern part of Manawatu Basin. These locations mostly presented with young sediments preserved by the Manawatu River, which formed a range of plain terraces. The second lowest has values ranging from 2 to 5 and is characterized by more hilly areas formed by some smaller river systems and older alluvial sediments from Miocene–Pliocene periods, which are the main sources for the Manawatu. Its locations spread in the northwestern and eastern parts of the basin. Geodiversity values 5–8 are presented only in the mountain ranges from the north to the south or around these formations, as well as small areas on the east; they are more connected with the oldest Mesozoic greywacke. Finally, the high and highest values contain 8–12 and 12–35 points, respectively, described together as they cover small areas mostly concentrated in the high areas of the mountain ranges and in the west part of Manawatu Basin. Specifically, the highest values are only concentrated in the Manawatu Gorge presented with some basaltic sequences and some more towards the southwest of the Tararua Range. Hence, the area of the Manawatu Basin is mostly presented with low and the lowest geodiversity values based on geomorphological and geological elements. Meanwhile, the high and highest values are mostly concentrated in the mountain ranges and the Manawatu Gorge.

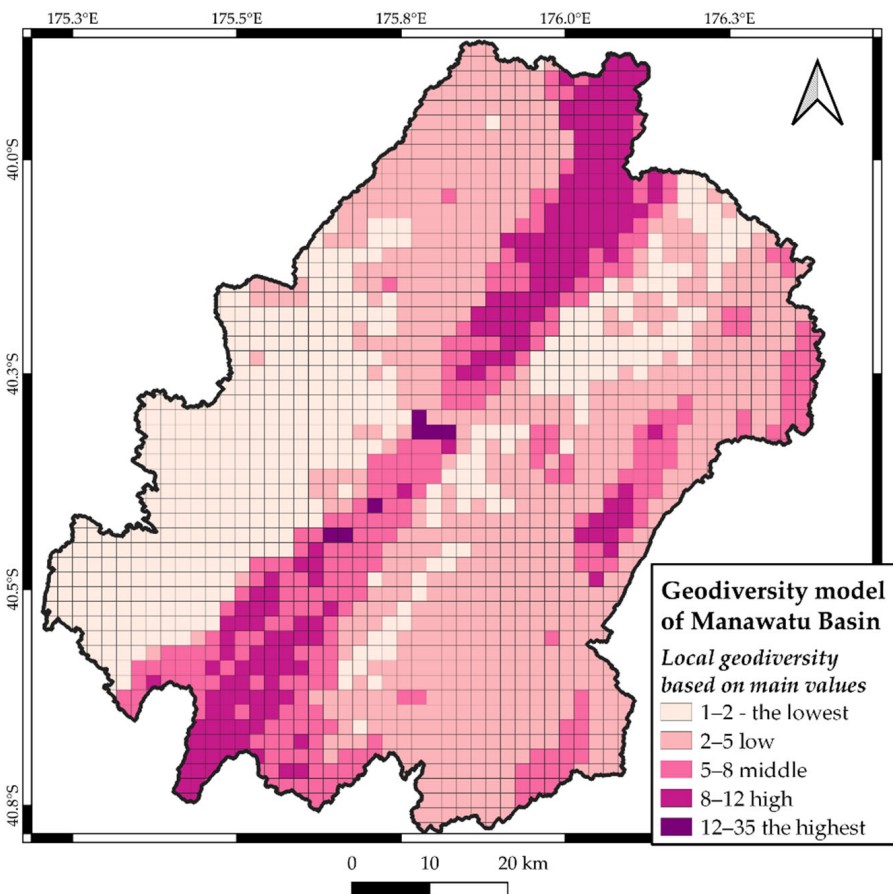

**Figure 6.** Geodiversity model of Manawatu Basin. Values were weighted specifically for local perspectives to highlight the locations of interest.

To improve the QQG assessment for geosite recognition into the result of local geodiversity, we added the parameter of Strahler order (Figure 7), which is based on the same SRTM data previously utilized for calculation of the slope model. The "Strahler order" model has been analyzed with the accordingly named tool of "Terrain Analysis-Channels" proposed by the Saga GIS plugin in the QGIS software. The model is added on top of the previously described geodiversity model of the Manawatu region. The result of calculations shows that a range of important changes occurred along the main streams and rivers with a high range for Strahler order, which highly influenced the final model. First, the range for all values is increased by the natural breakers mode, which simultaneously triggers a decrease in the number for areas with the lowest values for geodiversity. This has been provoked by the southwestern or lower part of the Manawatu River, which is flowing through plain areas and has the highest value for Strahler order so obtains a high value for local geodiversity by functioning as a major collector of the greatest variety of rocks sampled. Then, some areas with low values are raised to the middle range along tributary rivers flowing to the main flow from the northeastern part of the Ruahine Range and from the eastern part of the Manawatu Basin collecting in the Manawatu Gorge. Meanwhile, except for the lower part of the Manawatu River, most locations with the high and the highest values remain unchangeable, especially in Manawatu Gorge and in the northwestern part of Tararua Range. Hence, the Strahler order has a significant influence on the general geodiversity model, especially for places with otherwise low values. In areas along the main flow channel, it became an important additional parameter to elevate the local geodiversity value. However, it has a small impact on geodiversity values in areas where the general geodiversity is high.

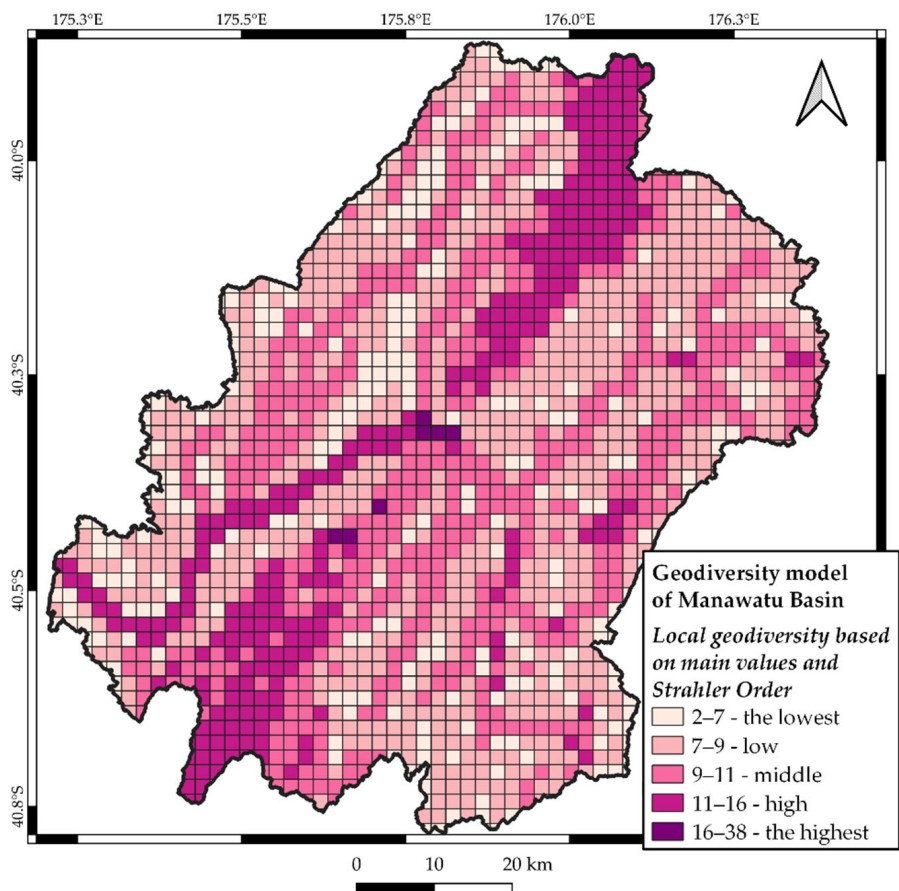

**Figure 7.** Geodiversity model of Manawatu Basin improved with hydrological values based on the model of Strahler order. Values were weighted specifically for local perspectives to highlight the locations of interest.

## 5. Discussion

The aim of this research is to demonstrate the influence of Strahler order on the qualitative–quantitative assessment of geodiversity, which based on geological and geomorphological elements describing the core parameter of abiotic nature. The result of the assessment shows that Strahler order mostly influenced the places with low and middle values for local geodiversity. In this assessment, we cannot consider global geodiversity for hydrological description of the Manawatu Basin because we should make an objective calculation of Strahler order through the Earth's surface separately for each basin to find out the highest possible value to create an evaluation system for the hydrological aspect. Meanwhile, from a global perspective the Manawatu region is considered as a place with low and the lowest values for geodiversity, which can contain some specifically important locations (for example Manawatu Gorge); however, they have to be studied more accurately with larger amount of data, which requires more time and resources. Hence, here we demonstrate local geodiversity values for the Manawatu Basin, where the Manawatu River is the main flow. It compounds in the western part from (the number) rivers merged into one flow up to the Manawatu Gorge. Therefore, all these streams contain values for Strahler order, which objectively increased the geodiversity values for the region with its terraces, especially in the lower part of the Manawatu River flowing from the Manawatu Gorge towards the southwest and falling into Tasman Sea. This part of the river raised the value of the area from the lowest to the high values. Meanwhile, the high and highest values remain on relatively the same areas. On the other hand, not all places get increased; the greywacke formation of the Waewaepa Range in the eastern part of the basin decreased the area of high values. This result was influenced by the natural breaks mode, which

merged it with high values to the surrounding area (middle values). In conclusion, hydrological element calculated as the Strahler order must be included into the assessment, as it increases the importance of the places with geologically and geomorphologically low values and does not highly influence the places with the highest geological values. This phenomenon standardly correlates with the natural law of water to gather in the basins and lowlands, creating terraces of alluvial deposits along its flow. Hence, general geological and geomorphological elements work better for volcanic and metamorphic rocks with steep slopes, whereas hydrological elements presented with the Strahler order improves the opposite flatland and depression areas. In conclusion, the Strahler order is a good parameter to include into QQG assessment as it increases the value of geodiversity in lowlands, which makes a more complete region with possible geosites in the Manawatu region; however, this still requires more accurate study in the field.

The Strahler order as a hydrological implication of general geodiversity (geology and geomorphology) contains several issues. First, the poor accuracy of calculated channels, which often take completely different directions than in reality or on the 1:50,000 New Zealand topographic map (https://www.linz.govt.nz/products-services/maps/new-zealand-topographic-maps, accessed on 19 December 2022). Meanwhile, the Strahler order accuracy is not dependent on the quality of DEM, as the same result is given for SRTM (30 m pixel) and DEM from the topographic map (8 m pixel). Next, the issue is that too many streams are calculated from the surface, where most of the time all channels from order one to around four are not real streams. However, we have left this drawback in the calculation, as to solve this problem the researcher needs to go through checking of all streams, deleting and correcting according to the available topographic map and satellite image. Additionally, low channel orders represent catchment areas that can be filled temporarily with precipitation. Hence, the hydrological model has not been changed as it still shows actual and potential water channels in the Manawatu Basin. The final issue is the result itself, which shows the whole river as important place; therefore, the researcher still needs to go along the whole stream on the field trip or at least utilize maps to highlight specific locations (possible geosites). In conclusion, despite all the problems described above, the assessment of Strahler order is fit for the aim of QQG assessment to minimize the areas for the search of geosites that still require more accurate checking of locations with high and the highest values. However, it must be improved with additional data about geodiversity and its accessibility.

The next possible step for improvements in geosite recognition in the Manawatu region is to apply more data about geodiversity elements, which can be presented as some specific features and/or locations with significances in a range of disciplines such as science, culture, history, esthetical, and many others [21,22]. Specifically, for the Manawatu region the LINZ database provides a decent number of locations that can be considered during the next field observation. These objects are historic sites and Māori Pa, which have cultural and historical significance; then rock outcrops, caves, and waterfalls that are mostly important for natural science and esthetics (Figure 8). Meanwhile, with a geosite search along the streams, the LINZ data provides historical locations and a Māori Pā (fortress) located along the lower Manawatu. Next, the waterfalls can be studied in the Tararua Range in the south and southeast of the Manawatu Basin itself and also the one in the Ruahine Range. Meanwhile, in the northern part of the basin, one cave and an outcrop can be found as well, which already makes this region scientifically important. Other outcrops can be found in the place with high value in the east side, as well as a final one in the far south in the already mentioned Tararua range. Therefore, this data has already improved the value for geodiversity in the Manawatu Basin, where flat areas with rivers contain places with historical and cultural values, whereas mountains have more scientific significance. In conclusion, future assessment requires improvement of the current geodiversity with additional data for establishment of locations with particular significance.

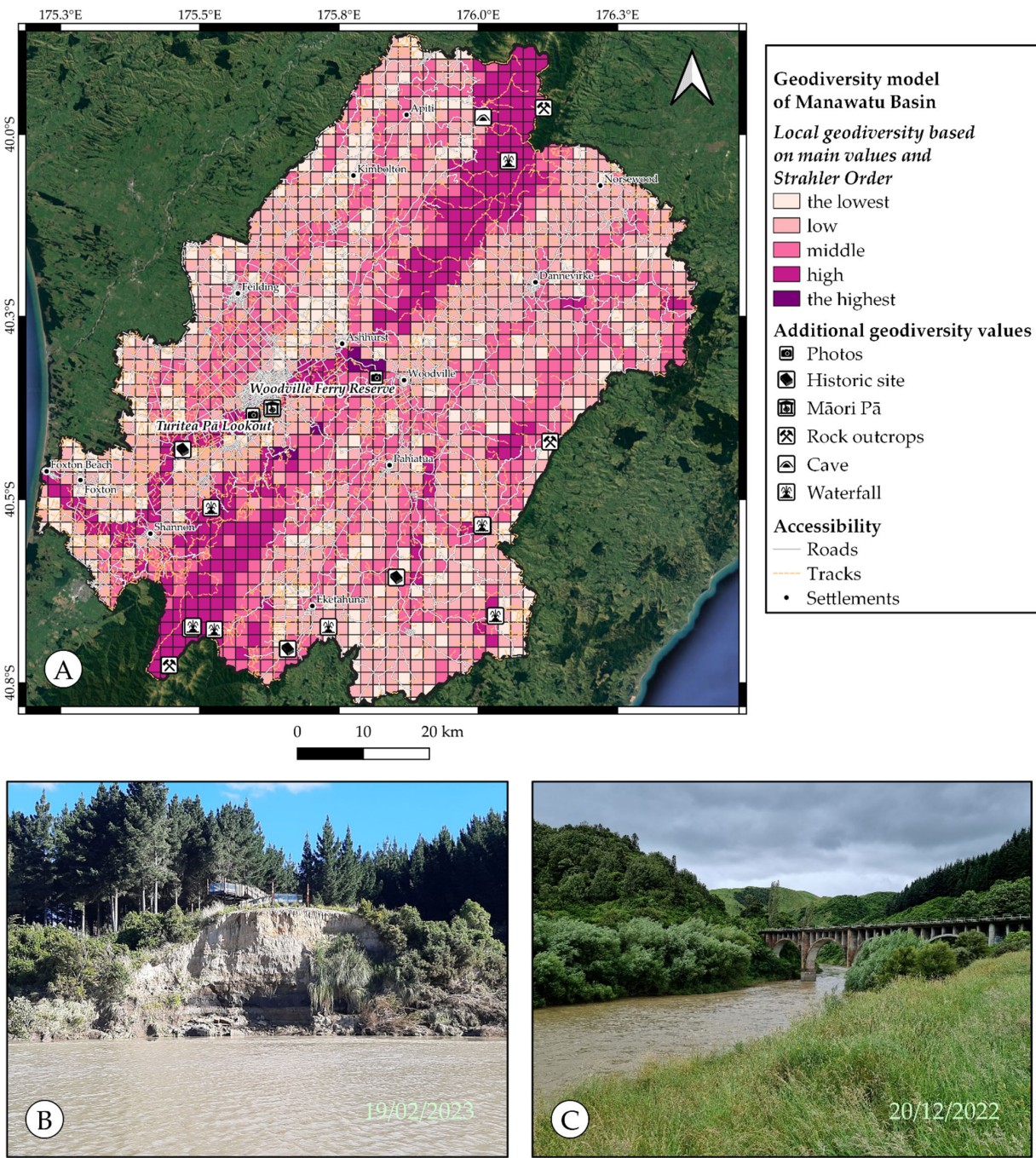

**Figure 8.** (**A**) Geodiversity model of Manawatu Basin with hydrological model and additional locations of geodiversity significance. Values were weighted specifically for local perspectives to highlight the locations of interest; (**B**) Turitea Pā Lookout; (**C**) Woodville Ferry Reserve.

The role of rivers in geoheritage characterization and geodiversity estimates is still underutilized. There are only a few exceptions of recent works where the geological and geomorphological aspects of rivers are proposed to apply to a more widespread sense. These works, however, are very specific to major valley systems and their geology and landscape elements within high mountain regions such as those in Kashmir [56,57]. Commonly, rivers are treated as zones along specific recognized geoheritage values such as the special rock formations along the Mekong River in Thailand; however, the focus of that research was on the rock formations and not on the role of the river in contribution to the geoheritage and geodiversity of the region [58,59]. A probably comparable approach

to explore the role of the river in the geoheritage scene was demonstrated and how the river itself can contribute significantly to elevating the overall geoheritage value of a region was shown for the Belaya River in SW Russia, where other geoheritage values are not as obvious [60]. As our research also showed, rivers are important elements of the overall geoheritage, and they commonly function as well-defined regions with significant geocultural values as well as dramatic scenery that can have geotouristic value [61,62]. There are very few works exploring the potential destruction of river systems from a geoheritage perspective, despite the rapid urbanization that can alter the natural geological and geomorphological features including raw material exploitation such as is the case for the River Nile along the greater Cairo region [63,64]. On the bright side, there are good initiatives to specifically categorize fluvial- and hydrological-process-related heritage termed as geo-hydrological heritage within their specific sites [65]. Also, the recognition of the geoheritage of rivers in geotourism and other niche tourism perspectives is a rapidly growing field and a promising direction for sustainable development [66,67].

Additionally, we provide some update to demonstrate catastrophic geomorphological changes, which happened to the Lower Manawatu River after flooding due to the high precipitation of cyclone "Gabrielle". The Category 3 cyclone Gabrielle hit the North Island of New Zealand, causing extensive damage mostly in the Northland, Auckland, Coromandel, and Hawke's Bay areas. The cyclone formed on the 6 February 2023 and had largely dissipated by the 16 February 2023. Alongside the destruction in the narrow core of the storm, extensive intense rainfall affected much of the territory of the North Island, including the Manawatu River region. Although there was less damage in the Manawatu Basin in comparison to other regions within the path of the cyclone, it generated intense rainfall that triggered flooding across the country. The overflow of the Lower Manawatu River channels happened through collection of most streams from the north coming from the Ruahine Range. Hence, the Manawatu River overflowed its banks, especially the true right side around Palmerston North (Figure 9). This demonstrates the rate of changes for the river in the area with the highest Strahler order (Figure 3). This flood event is comparable to the major flood recorded in 2010 (Figure 9). This flood demonstrates the rate of influences from the northern streams during high-intensity rain events.

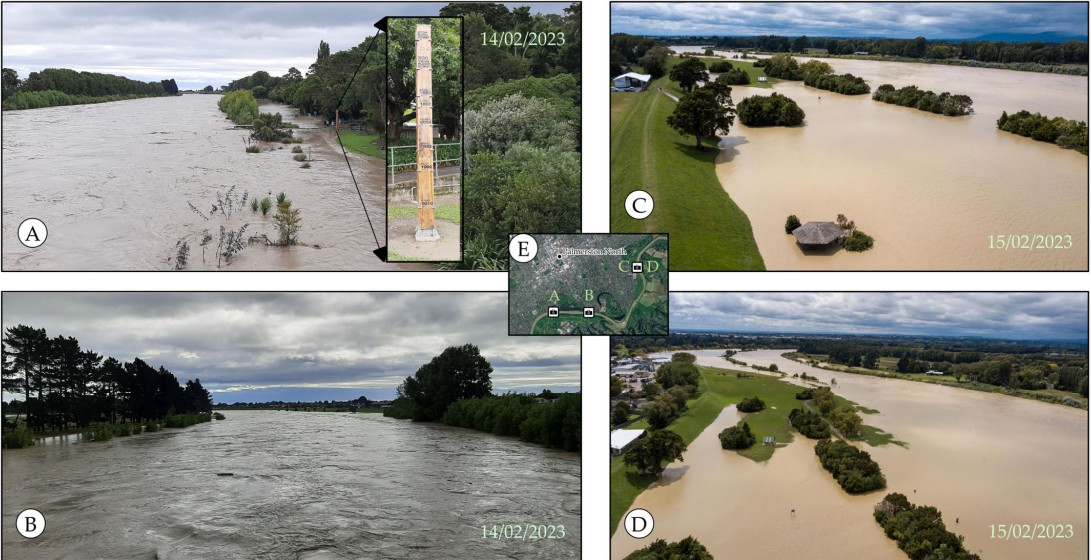

**Figure 9.** Photos on the topic of a flood that happened in the middle of February 2023. (**A**) Fitzherbert Bridge, Palmerston North; (**B**) He Ara Kotahi (can be compared with Figure 4), Palmerston North; (**C,D**) upstream Manawatu near the Higgins Industrial Area, Palmerston; (Photos (**C,D**) were taken by Matthew Irwin (Massey University, Palmerston North, New Zealand). (**E**) Overview map of locations, where photos have been taken.

## 6. Conclusions

Hydrological data expressed as Strahler order has a decent and positive influence on QQG assessment of geodiversity. It has an impact on the places with low and the lowest values as hydrological data fills all kinds of valleys and depressions throughout the area of research. Hence, the areas with low values have been increased to high. Particularly, this can be seen in locations along the Lower Manawatu. Hence, the Strahler order has a positive effect on the QQG assessment, especially for locations with evolved river systems.

The Strahler order model has number of issues during assessment, with inaccuracy compared with reality and the topographic map being the main drawback. Moreover, a high number of channels can be neglected even though most of them are potential channels; they are considered as potential drainage systems activated during high precipitation. The last issue is a result that highlights the whole river as an important place that still must be studied further to select the most significant parts. However, it is a consideration for the future research for geosite description, whereas it still fits the aim of this manuscript. Hence, the Strahler order model has several issues that can partly be solved through more precise correction, whereas others can be neglected as they do not influence the aim of research.

The information from the LINZ database provides a better picture of the geodiversity of the Manwatu Basin, which must be considered for future research to be included into QQG assessment of geodiversity as additional values along with hydrology. Cultural significance provides data about historical sites and Māori Pā, whereas data regarding rock outcrops, waterfalls, and caves can be considered important for geotourism and geoeducation. Hence, additional information about the uniqueness of the Manawatu Basin can significantly increase its geodiversity value.

**Author Contributions:** Conceptualization, V.Z.; methodology, V.Z.; software, V.Z.; validation, K.N.; formal analysis, V.Z.; investigation, V.Z.; resources, V.Z.; data curation, K.N.; writing—original draft preparation, V.Z.; writing—review and editing, K.N.; visualization, V.Z.; supervision, K.N.; project administration, K.N.; funding acquisition, K.N. All authors have read and agreed to the published version of the manuscript.

**Funding:** This research was funded by Massey University Post-Graduate Research Scholarship granted to V.Z.

**Data Availability Statement:** Not applicable.

**Acknowledgments:** This research is part of VZ's PhD research on the Manawatu basin funded by the Massey University PhD Scholarship. Thanks to Matthew Irwin (Massey University, Palmerston North, New Zealand) for photos of Manawatu River during flood incident provided to the manuscript.

**Conflicts of Interest:** The authors declare no conflict of interest.

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
