# Peer review of "Recognition of Potential Geosites Utilizing a Hydrological Model within Qualitative–Quantitative Assessment of Geodiversity in the Manawatu River Catchment, New Zealand"

_geographies, doi:10.3390/geographies3010011_

Round 1

Reviewer 1 Report

A very interesting article that puts another brick in the great construction of the important topic of geodiversity, both in qualitative and quantitative terms.

The manuscript is structured as required by international scientific standards, the research objective is well focused, the methods adopted are consistent with the objective of the paper, and the results and discussion are balanced and fair. The figures are attractive and well-edited.

This draft article is therefore worthy of publication in the journal Geographies.

I only ask the authors to write a few more lines about the presented case study, its importance to the scientific community and whether upscaling of the result is possible.

Some considerations:

Fig. 1 what does the “geomorphology “ of the box on the right mean? It is not a geomorphological map.

Fig. 2 is it possible to include the type of rock (sandstone, shale, siltstone, etc.) in the legend?

Fig. 5 what part of the basin is represented?

Then there are some articles on geodiversity in the MDPI journals that could probably provide some additional useful insights, e.g.

de la Hera-Portillo, Á. et al. Geodiversity of Las Loras UNESCO Global Geopark: Hydrogeological Significance of Groundwater and Landscape Interaction and Conceptual Model of Functioning. Resources 202312, 14. https://doi.org/10.3390/resources12010014

Ferrando, A. et al. A Quantitative GIS and AHP Based Analysis for Geodiversity Assessment and Mapping. Sustainability 202113, 10376. https://doi.org/10.3390/su131810376

Perotti, L. et al. M. Geodiversity Evaluation and Water Resources in the Sesia Val Grande UNESCO Geopark (Italy). Water 201911, 2102. https://doi.org/10.3390/w11102102

…and others

Reviewer 2 Report

Dear Authors,

I’ve read your manuscript with great interest. I belong to those people who have always argued that hydrological objects should be treated together with geoheritage and geodiversity. So, I’m happy to see contribution sharing these views. The paper is based on innovative, novel research, the outcomes of which will be important to the international research community. It is methodologically strong and informative. It is also well-written, well-structured, and well-illustrated. I value this work highly, and only minor, easy-to-follow amendments can be recommended.

1)      Key words: Cenozoic and Mesozoic sediments -> Mesozoic; Cenozoic

2)      Introduction: please, cite some previous works where hydrological heritage and hydrological geosites were considered. Particularly, these papers may be cited I guess:

https://www.cell.com/heliyon/fulltext/S2405-8440(22)03698-2

https://www.mdpi.com/1660-4601/20/1/565

https://www.tandfonline.com/doi/abs/10.1080/00167487.2006.12094145

https://www.tandfonline.com/doi/abs/10.1080/08873631.2013.828482

3)       Subsection 2.1: please, polish geological descriptions. I may be wrong, but I feel a mix of different terms. For instance, terrane is a tectonic term, whereas you characterize stratigraphy; greywacke is a compositional term, which can be applied to both sandstones and siltstones, etc. You write about Miocene-Pliocene sediments: well, it is possible to write so in some languages, including my own Russian. However, some foreign colleagues have urged that sediments can be only Quaternary in age, whereas all older deposits, if even soft, are rocks, and, thus, the word “deposits” is more appropriate.

4)      Subsection 3.1: I’d recommend replacing “qualitative-quantitative assessment” with simply “assessment”.

5)      Line 347: what are jenks?

6)      Discussion: you should cite some sources. Please, try to link your study to the research of the others.

7)      I see a nice bridge on Figure 8. Although this is artificial element, it may influence on the geodiversity value, isn’t it?

8)      Figures: please, avoid the label “Legend” on the figures.

9)      Figures: when they are composite and include several drawings/images, please, label each as A, B… and explain each item in the caption.

Good luck with revisions!

Reviewer 3 Report

Gaps have not been identified properly.
